# Spectral Domain Optical Coherence Tomography Findings in Vision-Threatening Rhino-Orbital Cerebral Mucor Mycosis—A Prospective Analysis

**DOI:** 10.3390/diagnostics12123098

**Published:** 2022-12-08

**Authors:** Ankur Singh, Preeti Diwaker, Akanksha Agrawal, Aniruddha Agarwal, Jolly Rohatgi, Ramandeep Singh, Gopal Krushna Das, Pramod Kumar Sahoo, Vinod Kumar Arora

**Affiliations:** 1Department of Ophthalmology, University College of Medical Sciences and Guru Teg Bahadur Hospital, Delhi 110095, India; 2Department of Pathology, University College of Medical Sciences and Guru Teg Bahadur Hospital, Delhi 110095, India; 3Cleveland Clinic Lerner College of Medicine, Case Western Reserve University, Cleveland, OH 44106, USA; 4Vitreoretina and Uveitis Service, Advanced Eye Centre, Post Graduate Institute of Medical Education and Research, Sector 12, Chandigarh 160012, India

**Keywords:** rhino-orbital mucormycosis, optical coherence tomography, retinitis, central retinal artery occlusion, ophthalmic artery occlusion, disruption of retinal layers

## Abstract

Rhino-orbital cerebral mucor mycosis is a rare disease entity, where retinal involvement is described in the literature mostly as CRAO. However, pathological studies have shown mucor invading the choroid and retina with a neutrophilic reaction. So, it is pertinent that retinal inflammation secondary to invading mucor has some role in microstructural changes seen in the vitreous and retina of these patients. This novel study aims to describe the vitreal and retinal features of patients with vision-threatening rhino-orbital cerebral mucor mycosis and how they evolve on spectral domain optical coherence tomography (SD-OCT). This study shall also provide insight into the pathophysiology of these vitreoretinal manifestations by in vitro analysis of the exenterated orbital content. Fifteen eyes of fifteen patients with vision-threatening ROCM treated with standard care were enrolled in this study and underwent complete ophthalmic examination, serial colour fundus photography, and SD-OCT for both qualitative and quantitative analysis, at baseline and follow-up visits. SD-OCT on serial follow-up revealed thickening and increased inner-retinal reflectivity at presentation followed by thinning of both, other features such as the loss of the inner-retinal organized layer structure, external limiting membrane (ELM) disruption, necrotic spaces in the outer retina, and hyperreflective foci. Vitreous cells with vitreous haze were also seen. There was a significant reduction in CMT, inner and outer retinal thickness, total retinal thickness (all *p* < 0.05) with time, the quantum of reduction concentrated primarily to the inner retina. In summary, in vivo and in vitro analysis revealed that early microstructural changes were primarily a result of retinal infarctions secondary to thrombotic angioinvasion. With the late microstructural changes, there was possible sequelae of retinal infarction with some contribution from the inflammation, resulting from mucor invading the choroid and retina.

## 1. Introduction

Rhino orbital cerebral mucormycosis (ROCM) is a highly morbid and rare disease caused by saprophytic fungi belonging to the genera Mucor, Rhizopus, and Absidia belonging to the Mucoracea family [1]. The risk factors predisposing patients to mucor-mycosis are uncontrolled diabetes, neutropenia, haematological malignancies, organ transplantation, trauma and burn, and use of immunosuppressants such as corticosteroids [2]. Post COVID-19 pandemic, the Asian sub-continent has faced the largest upsurge of ROCM, in a varied presentation with most cases being detected in India [3]. This upsurge can largely be attributed to the high prevalence of diabetes mellitus and the use of dexamethasone in the treatment of COVID-19 infection [3].

Unless diagnosed and treated early, ROCM has a poor prognosis and high mortality rates. Inhalation of spores into the nasal or oral cavity is the primary route of infection in cases of ROCM. It is from here they make their way into sinuses and then through lamina papyracea, inferior orbital fissure or orbital apex to the orbital cavity. The intracranial extension may happen from orbit or hematogenous dissemination from the lungs leading to a high rate of fatality (49%) [3]. An angioinvasive nature and resultant occlusion of the vessel make treatment of ROCM difficult, requiring aggressive early surgical debridement along with local and systemic injections of antifungals such as amphotericin B. Early surgical intervention includes debridement of infected tissue within the paranasal sinus, at times combined with orbital exenteration [4]. Early cerebral involvement along with deranged metabolic parameters due to pre-existing systemic conditions and toxicity of systemic antifungals makes aggressive surgical management difficult and may be associated with the high fatality of the disease.

Ocular involvement in cases of ROCM initially presents as predominantly orbital cellulitis with ophthalmoplegia in almost 80% of cases [3]. Visual loss occurring in these cases can be attributed primarily to ocular and optic nerve ischemia, central retinal and ciliary artery occlusion resulting from fungal infiltration of blood vessels, optic apex syndrome, keratitis, retinitis and, in a few cases, endophthalmitis [5,6,7]. Histopathological analysis of these eyes has shown thrombosing arteritis of both retinal and choroidal vasculature with a lesser extent of venous involvement. Mucor hyphae were also seen to invade nerves, muscle, sclera and bone, and have been demonstrated in all ocular tissue except lens and vitreous [8]. The intraocular tissue reactions in these patients range from none to that of acute pan-ophthalmitis, requiring a tailored treatment approach.

Multimodal imaging has become an integral part of ophthalmologists’ standard clinical armamentarium and is considered useful for the diagnosis and management of various retinal diseases and uveitis [9,10]. Few case reports have described retinal and vitreous manifestation of orbital mucormycosis on colour fundus photography, fundus fluorescein angiography (FFA), but the description is limited to central retinal artery occlusion, endophthalmitis and a single case description of chorioretinal alterations [11,12,13,14,15]. No study to the best of our knowledge has prospectively analysed vitreoretinal features in rhino-orbital mucormycosis (ROM) or ROCM on spectral-domain optical coherence tomography (SD-OCT). SD-OCT being a rapid, noninvasive and a repeatable imaging modality that provides near histological resolution images of the posterior segment of the eye. Being non-invasive and rapid, the diagnostic tool is promising and effective also in critically ill patients with multi-organ dysfunction. The index study aims to describe the vitreal and retinal findings in patients affected with ROM/ROCM in various stages of the disease, covering the complete spectrum of retinal and vitreous manifestation of disease on SD-OCT and to draw a clinicopathological correlation. This new information provided would not only be helpful to improve the understanding of the pathophysiology of the disease, but may also aid in diagnosis and management and follow-up of this rare entity.

## 2. Materials and Methods

Those prospectively enrolled in the study were microbiologically, radiologically and histopathologically proven consecutive patients of ROM/ROCM with orbital involvement, admitted at the tertiary referring center in North India (Guru Teg Bahadur Hospital, New Delhi) between June 2021 and August 2021. They had recent onset decreasing or loss of vison secondary to posterior segment pathology, and not attributable to the ocular manifestation of underlying systemic disease. The preregistered study protocol can be found on the Clinical Trials Registry—India (CTRI/2022/05/042370). Institutional Review Board/Ethics Committee approval was obtained for the protocol before the conduct of the study. The study adhered to the tenets of the Declaration of Helsinki and the rules laid down by the Health Insurance Portability and Accountability Act of 1996. A written informed consent was obtained from all participants at the start of study.

The diagnosis of mucormycosis was based on microbiological confirmation for mucorales done with direct microscopic preparations, made with 15% potassium hydroxide, of the samples drawn from deep nasal swabs or endoscopic guided nasal swabs or nasal biopsy. Orbital involvement was confirmed by contrast-enhanced computerized tomography or Gadolinium magnetic resonance imaging for the presence of radiological signs suggestive of orbital mucormycosis. Posterior segment involvement was judged by funduscopic examination by an expert retina consultant. The inclusion criteria for this study were (a) microbiologically and histopathologically confirmed cases of ROM/ROCM with radiological imaging suggestive of invasive fungal sinusitis and orbital involvement, (b) retino-choroidal involvement not explained by any other systemic disease, and (c) a normal carotid doppler and echocardiography study. The exclusion criteria were (a) posterior segment involvement as an ocular manifestation of an underlying systemic disease (such as diabetic retinopathy, hypertensive retinopathy, or its complication/sequelae) or medical records suggestive of any pre-existing retino-choroidal pathology (such as old CRVO, healed posterior uveitis, or treated retinal detachment), (b) presence of media opacity that precludes a clear view of the fundus on fundus photography and well adversely affected of OCT scans accusation, and (c) patients not consenting for study. Past and recent medical records including history of COVID and steroid use were documented for all the subjects included in the study. All recruited patients underwent a complete ophthalmoscopic examination at baseline visit, including best-corrected visual acuity, slit lamp examination, intraocular pressure measurement (Goldmann Applanation Tonometer), and funduscopic examination (both with slit lamp and with indirect ophthalmoscope).

All patients with proven ROM/ROCM and orbital involvement were started on induction therapy of intravenous liposomal amphotericin B 5–10 mg/kg body weight or intravenous posaconazole 300 mg twice on the first day followed by 300 mg once a day onwards in patients with impaired renal functions. Aggressive surgical debridement of paranasal sinus with clear margins was carried out by ENT specialists immediately on admission or as early as possible. Patients with orbital involvement on CT/MRI were also treated with transcutaneous retrobulbar amphotericin B 3.5 mg/mL injections. Patients with central nervous system (CNS) involvement were treated in concordance with neurosurgeons. Patients with absolute loss of vision (no perception of light), extensive orbital involvement and loss of enhancement of orbital tissue on MRI underwent orbital exenteration. The exenterated orbital content was sent for histopathological evaluation. All cases were put on at least four weeks of intravenous liposomal amphotericin B until clinical regression, radiological regression, or stabilisation and reconstitution of the host immune system was achieved. This was followed by step-down treatment of oral posaconazole 300 mg twice on the first day followed by once a day for at least 3–6 months. During the course of treatment, strict metabolic control of patients was maintained by an internist.

Patients with posterior segment involvement at baseline were imaged using colour fundus photography and SD-OCT (Retina Scan-Duo, NIDEK Co., Ltd., Gamagori, Japan) and then prospectively on a weekly basis for a total duration of induction therapy, or two months, whichever was later. Patients undergoing orbital exenteration were imaged a day prior to orbital exenteration where ever possible.

### 2.1. Demographic and Clinical Profile of Enrolled Patients

Fifteen eyes from fifteen patients (twelve males and three females) fulfilled the inclusion criteria and were enrolled in the study. The mean age of the subjects was 46.5 ± 13.2 (range: 32–66) years. Thirteen patients had diabetes mellites as an underlying systemic condition, with seven patients having a history of steroid intake as a treatment for COVID-19-associated pulmonary manifestation. The patients’ details, including medical history, extraocular findings, follow-up duration, and treatments received, are summarised in Table 1 and Table 2. Among the fifteen patients enrolled, four underwent orbital exenteration during the course of treatment with a mean duration of follow-up 4 ± 2.9 weeks. The mean follow-up of the remaining eleven eyes was 8.6 ± 1.02 weeks.

### 2.2. Image Acquisition

The SD-OCT scans were performed with SD-OCT (Retina Scan-Duo, NIDEK Co., Ltd.) at a scan rate of 13,250 A-scans/second (Ultrafine mode) using macula line protocol of 12 mm scan size (length) composed of 1024 equally spaced transverse sampled locations centred on fovea. Auto tracking and auto shot utility were used to acquire images to minimize motion artifacts and enhance image comparability over time. In the eye, where auto tracking and auto shot were not possible due to ocular motility restriction, manual 12 mm foveal scans centred at fovea were acquired. The scan position was confirmed for comparability on follow ups before comparing the serial images. For macular thickness, a macular map protocol consisting of 128 horizontally oriented B-scans, each 9 mm in length and composed of 512 equally spaced transverse sampled locations, was used. All 128 OCT B-scans were acquired in a continuous, automated sequence and cover a 9 mm × 9 mm area. A circular topographic macular map of three circular areas (including the foveal region, an inner macular ring, and an outer macular ring) with early treatment diabetic retinopathy study-type sectors, was generated for obtaining the central macular thickness. Modified 7-field 30° colour fundus photographs were obtained for all the subjects at baseline and follow-ups using in-built panorama mode. (Retina Scan-Duo, NIDEK Co., Ltd.). 12 mm and 9 mm line scans were also acquired centred over the area of interest wherever they were required.

### 2.3. Image Evaluation

Vitreal and retinal findings for structural changes were analysed on 12 mm and 9 mm foveal line scans were acquired. Choroidal findings for structural changes were also analysed wherever possible. Central macular thickness (CMT) was measured within the innermost central circle (1000 mm in diameter) of the macular thickness scan. Total retinal thickness and the thickness of the inner and outer retinal layers were measured on foveal line scans at 1 mm nasal to fovea. Total retinal thickness was calculated between ILM-RPE using an inbuilt thickness graph. The inner retinal thickness, consisting of the combined internal limiting membrane, retinal nerve fibre layer, ganglion cell layer, inner plexiform layer, and inner nuclear layer in the macular area, was measured manually using inbuilt callipers of the OCT system. For structural assessment the outer retina was defined as the combined outer plexiform layer, outer nuclear layer, inner and outer segment, and RPE. In patients with unilateral involvement, retinal findings were compared with the contralateral eye for structural integrity of the retinal layers and changes in the thickness or reflectivity of specific retinal layers at baseline. All SD-OCT images were reviewed by two independent experienced observers (A.S. and A.A.) for qualitative as well as quantitative analysis. In case of disagreement, a third operator (R.S.) was asked to judge the images. The mean values between the two operators was used for statistical analysis.

### 2.4. Histopathology

The exenterated eyes were immediately placed in 10% buffered formalin. The exenteration specimens were kept for overnight fixation in the histopathology laboratory after bisecting the eyeball in horizontal pupil-optic nerve plane. Cross section of the optic nerve, and sections incorporating the macular region along with the optic nerve, were taken. The tissue sections taken were processed for routine histopathological examination and the slides were stained with haematoxylin and eosin (H&E) stain. Sections from the macular region were reviewed by three pathologists who were blinded to each other’s findings. In case of disagreement, the slides were reviewed by all three pathologists together and a common consensus was taken as final.

### 2.5. Statistical Analysis

Statistical analysis was performed using GraphPad Prism (GraphPad Software Inc., La Jolla, CA, USA). The differences in the mean inner, outer retinal, and total retinal thickness at 1 mm nasal to fovea and CMT at baseline examination, at one month and at two months, were calculated using the mixed model. The differences in the mean inner, outer retinal, and total retinal thickness at 1 mm nasal to fovea and CMT of affected eyes to that of contralateral control eyes were calculated individually, at baseline examination, at one month and at two months, using an unpaired *t*-test. Visual acuity measurements were converted to the logarithm of the minimal angle of resolution (LogMAR) for all analyses. Paired *t*-test was used for baseline and final visit BCVA. The level of significance was set at 0.05.

## 3. Results

The prevalence of OCT findings at baseline, one month and last follow-up in the affected eye are summarised in Table 3.

### 3.1. Morphological Features on OCT at Baseline

At presentation, assessment of retinal and vitreous microstructure changes was possible for all fifteen eyes. Five of these eyes, although with minimal intensity, showed a peculiar localised increased background hyperreflectivity in posterior vitreous (vitreous haze), localised mostly in the peripapillary region. Hyperreflective dots (vitreous cell) in posterior vitreous were seen in ten of the eyes overlaying the disc in most of the cases. This accompanied posterior hyaloid thickening with localised detachment in peripapillary area in one eye. Increased internal limiting membrane (ILM) reflectivity was observed in seven eyes in the peripapillary area with localised ILM detachment in five of the eyes. Inner retinal thickening was an observation common to fourteen (93.3%) eyes at presentation with associated increase in inner retinal reflectivity. Retinal folds, extending from juxtapapillary region to fovea, were observed in five eyes (Figure 1A). In all eyes with retinal folds, inner retina could not be differentiated from the outer retina. Increases in reflectivity of inner retinal layers made a differentiation of the hyperreflective inner retina from the outer plexiform layer difficult in three eyes (Figure 1B). It is only in seven eyes that inner and outer retina layer differentiation was possible for both qualitative and quantitative analysis. Loss of inner retinal organised layer structure along with outer plexiform layer was observed in thirteen (86.6%) eyes. Apart from inner retinal morphological alteration, outer retina was also seen to be affected in our cohort with increased outer retina thickness in four of the seven (57%) eyes at presentation. Other outer retinal alteration included external limiting membrane (ELM) disruption which was present in three of the seven eyes. In one eye, ELM disruption was seen along with photoreceptor layer disruption, seen as localised fuzzy disruption, of the ellipsoid layer in the juxta foveal region. The eye that showed photoreceptor layer disruption at presentation was delayed by a week for retinal imaging due significant facial and lower lid swelling secondary to early sinonasal debridement (Figure 2). In the same patient inner retinal thinning along with total, retinal thinning was also observed.

### 3.2. Morphological Features on OCT at One Month

In the first four weeks of follow-up, two patients, having nil visual acuity along with frozen globe at presentation, underwent orbital exenteration along with sinonasal debridement as a standard of care. Imaging for vitreoretinal morphological alteration revealed persistence of vitreous haze in four eyes. Though there was no increase in the prevalence of vitreous haze, an increase in intensity was observed in two eyes compared to that at baseline. Nine from thirteen eyes, although with variable frequency, revealed vitreous cells in the posterior vitreous. Two eyes among these had vitreous cells along with vitreous haze, overlaying the first branch of the central retinal artery. Although there was a significant decrease in inner retinal thickness in all eyes, inner retinal hyper-reflectivity persisted in 66.6% of eyes at the end of one month (Figure 3H). As observed during the presentation hyperreflective inner retina could not be differentiated from the outer plexiform in three eyes at the end of one month. In the other six from thirteen eyes, inner and outer retina layer differentiation was possible, with all these nine eyes showing loss of organised layer structure of the inner retina. There was a loss of organised layer structure for both inner and outer retina in four of the thirteen eyes, seen as hyper- or isoreflective-disorganised retina. At the end of four weeks, four eyes in our study developed ill-defined focal, optically empty spaces in the outer retina. In one patient, these optically empty spaces were observed in the inner retinal layer with a horizontal linear pattern, resembling the atrophic splitting of inner retinal layers. Few hyperreflective foci were visible in the outer retinal layer, although in a few other eyes they were seen along the inner and outer boundaries of the atrophic/necrotic optically empty spaces (Figure 1D). At the end of one month, there was an increase in the number of eyes showing ELM and photoreceptor layer disruption from baseline. This ELM and photoreceptor disruption was seen universally in all eyes, with increased intensity in the area between the optic disc and foveae. The retinal pigment epithelium (RPE), preserved in most eyes, was disrupted in one eye in the juxta papillary region.

### 3.3. Morphological Features on OCT at Last Follow-Up

As a part of treatment for ROCM, four eyes having extensive orbital involvement with posterior segment involvement underwent orbital exenteration. With the end of the induction phase of IV amphotericin B, by the end of seven weeks in most patients, there was a 5.6% (n = 2) reduction in the number of eyes showing vitreous cells, with vitreous haze persisting in four eyes, although with reduced intensity. There were increases in the number of eyes showing posterior hyaloid thickening with partial detachment over time, with three additional eyes developing posterior hyaloid thickening and partial detachment in the last month of follow-up. Somehow, increased focal ILM reflectivity was observed in 63.6% of eyes. There was a significant decrease in both inner retinal thickness and hyperreflectivity from that of baseline, with inner retinal hyperreflectivity persisting in two eyes at the end of two months. Large necrotic spaces seen at the end of one month in four eyes collapsed with time, leading to a further thinning of the retina (Figure 1H). At the end of two months, the necrotic spaces were reduced to trace with associated disorganised retinal layer structure. In all eyes except four, there was a loss of organised layer structure for all retina layers, with nontraceable or focally traceable external limiting membrane. Photoreceptor layer disruption was seen along with juxtapapillary loss of RPE and thinning of retina suggestive of scaring. A pattern was noticed in two eyes of possible contraction and clumping of the inner retinal layer with baring of the outer retina in the para-foveal area, suggestive of fibrous scarring of the inner retina. These eyes with scarring also had RPE hyperplasia, seen as clumping of the RPE layer with a shadowing effect. In three eyes there were also areas of focal chorioretinal scarring seen as disorganised, thinned out retina, along with loss of RPE layer and choroidal thinning. Retinal thinning was seen in 90.9% of eyes at the end of two months.

### 3.4. Changes in Quantifiable Parameters over Time

For quantifiable analysis, inner and outer retinal thickness was measurable in seven, six and four eyes at baseline, at one month and last follow-up, out of fifteen, thirteen, and eleven eyes, respectively (Table 3). At the baseline, the affected eyes had significantly greater CMT (*p* = 0.014) and total retinal (*p* = 0.040) as compared to the unaffected contralateral eye. Though the thickness of both the inner and outer retina was more in comparison to the contralateral unaffected eye at baseline, this difference was not significant for (inner retina *p* = 0.063), (outer retina *p* = 0.316) for both. Over time, there was a significant reduction in CMT (*p* = 0.010) and total retinal thickness (*p* = 0.001) inner (*p* = 0.0009) and outer retinal thickness (*p* = 0.008), with the quantum of reduction primarily concentrated on the inner retina (Table 3). At the end of two months, the affected eyes had significantly less CMT (*p* = 0.0003), total retinal (*p* < 0.0001), inner retinal (*p* = 0.003) and outer retinal thickness (*p* = 0.04) as compared to the unaffected contralateral eye. However, this difference in thickness metrics was not significant for both inner (*p* = 0.17) and outer retina (*p* = 0.42) at the end of one month. No significant difference was observed in VA at the presentation and at the last follow-up (Table 2).

### 3.5. Histopathology of Exenterated Eyes

The histopathological findings observed in the sections of four eyes with variable frequency included oedematous and degenerative changes in the optic nerve, choroidal and retinal tissue necrosis, disruption of the normal lamellar architecture of retina with the presence of neutrophilic infiltrate, presence of mucor hyphae in the necrotic tissue, thrombosed blood vessels and angioinvasion by fungal hyphae. Table 4 summarises the histopathological and last follow-up OCT findings of individual eyes undergoing orbital exenteration for vison-threatening ROCM.

## 4. Discussion

ROCM is a life-threatening infection, including 20–80% of patients having visual loss [16,17,18]. ROCM is associated with high residual morbidity and mortality due to the angioinvasive nature of fungus, thereby causing vascular occlusion and, consequently extensive tissue necrosis [19]. The vision loss in majority of patients have been attributed to optic nerve ischemia, ophthalmic artery occlusion, central retinal and ciliary artery occlusion [4,16,18]. Apart from angioinvasion, in vitro studies have found mucor, associated with intense polymorphonuclear reaction, in all coats of the eye. However, the prevalence of choroiditis, chorioretinitis and endophthalmitis in patients with orbital mucormycosis is limited to 1–6% [1,16,17,20]. Recent literature described retinal findings in patients with vision-threatening ROCM as central retinal artery occlusion (CRAO) and its sequelae for most cases. However, in view of histopathological evidence of mucor in the retina and choroid, it is pertinent that retinal inflammation due to mucor has some role in structural changes seen in the retina of patients. Spectral-domain optical coherence tomography can provide a quasi-histologic assessment of the posterior ocular fundus in vivo. SD-OCT recorded findings in the central retinal artery occlusion and different kinds of retinitis; choroiditis is well described in the literature [21,22]. Few anecdotal case reports have reported SD-OCT findings in patients with ROCM, but the description is limited to that of incomplete CRAO [23,24]. However, a detailed SD-OCT-based prospective analysis of vitreoretinal manifestation in patients of vision-threatening ROCM has not been elucidated. Infective affection of the retina and choroid and associated structural changes is also missing in published literature.

This index study identifies vitreoretinal structural changes in patients with vision-threatening ROCM on SD-OCT. With the aid of histology, this study will provide insight into the pathophysiology of retinal changes seen in patients with vision-threatening ROCM, which may aid in improving our understanding of these retinal manifestations. In our study, two events occurred concurrently; the ischemia caused occlusion of the retinal artery or ophthalmic artery due to the angioinvasive nature of mucor; and the inflammatory aftermath caused by the fungus itself in the retina and choroid. In this study, the fundus images of patients at presentation releveled diffuse retinal whiting of varying intensity, boxcarring of the retinal vessel, with or without cherry red spot, and in a few cases with papilla-macular fold suggestive diffuse retinal oedema. These findings were consistent with those seen in patients with central retinal or ophthalmic artery occlusion [22]. On SD-OCT, the patients in our study at presentation showed an increased inner retinal thickness of varying degrees of intensity, with inner retinal hyperreflectivity and loss of organized inner retinal layer structure. These findings were more prominent at the posterior pole for all patients. Schmidt and associates have categorized CRAO as incomplete, subtotal, or total CRAO subtypes based on the extent of retinal edema, degree of vision loss and delay in arterial blood flow [25]. The intensity of inner retinal oedema can be quantified with SD-OCT and has been seen to vary with the type of CRAO, ranging from mild in incomplete to severe in total CRAO [22]. Loss of organized inner retinal layer structure is also consistent with the severity of CRAO. Inner retinal hyperreflectivity as an acute event in CRAO is due to edematous opacification of the retinal nerve fiber and ganglion cell layer resulting from the cessation of axoplasmic transport after ischemic damage [26].

In the control arm, which is the non-affected eye at presentation in our study, the mean central macular thickness, inner retinal, and total retinal thickness at 1 mm nasal to fovea was (247.2 ± 13.7; 220–267) µm, (149.6 ± 9.6; 121–157.5) µm and (332.5 ± 20.4; 350–270) µm respectively which matched with the published normative data [27]. In our study at presentation, the affected eyes had a mean total retinal thickness at 1 mm nasal to fovea as (498.8 ± 300.2; 300–1080) µm, with five eyes having a total retinal thickness of more than 500 µm. In these five eyes on OCT, the inner retina could not be differentiated from the outer retina, and the clinical picture showed retinal folds, the absence of a cherry red spot with no light perception. These features suggest severe ischemic insult of the retina and choroid and can be attributed to ophthalmic artery occlusion [28]. Other features suggestive of total CRAO or ophthalmic artery occlusions were outer retinal oedema and inner retinal oedema seen in four eyes at presentation in our study [22,28]. In vitro analysis of an exenterated eye revealed thrombosis and angioinvasion of the central retinal artery along with the second-order retinal artery (Figure 4D,E). ILM detachment and increased focal ILM reflectivity as compared to the contralateral eye seen in a few eyes in this study at presentation could be attributed to early autolysis of the inner retinal layer seen as an acute event in CRAO [28]. The presence of ILM detachment and increased focal ILM reflectivity is suggestive of severe and persistent retinal hypoperfusion and is associated with poor final visual acuity. The eyes with ILM detachment at presentation tend to develop retinal thinning and atrophy of the inner retinal layers at the final follow-up [29].

Previous case studies have reported chronic features of CRAO on SD-OCT as decreased thickness and reflectivity of the inner retinal layers, decreased retinal thickness, and a corresponding increased reflectivity of the outer retina [30]. Ahn et al. reported outer retinal thinning and photoreceptor defects in cases of total CRAO and ophthalmic artery occlusion [22]. In our study, there was a reduction of 44% in eyes showing hyperreflectivity at one month, which reduced further by 16% at the end of two months. Inner retina hyperreflectivity has the highest significance for discriminating patients suffering from CRAO from normal subjects [30]. Reduction of inner retinal hyperreflectivity is usually a pseudo-normalisation of the inner retina as the hyperreflective edematous retina gets replaced by atrophic areas starting at as early as one month and becoming clearly atrophic at three months [26]. None of the eyes in our study showed increased retinal thickness at one month and two months. Outer retinal thinning was also observed in our study, with the quantum reduction in thickness more in the inner retina compared to the outer retina. Focal photoreceptor disruption was observed in all eyes from one month onwards, with disruption of ELM. These outer retinal alterations were more prominent at the posterior pole with relative sparing of the peripheral retina.

In this study, few vitreous cells and vitreous haze were observed in posterior vitreous, overlaying the disc, posterior pole and along the retinal vessel in 73.3% of eyes at presentation. Posterior hyaloid thickening and partial detachment was observed as late features. Vitreous cells, protein-rich vitreous haze, and posterior hyaloid thickening are features of posterior segment inflammation [31,32]. In vitro analysis of the exenterated eyes in our study showed mucor hyphae in the retina and choroid with traces of neutrophilic infiltrate (Figure 5C,D). However, even with in vitro evidence of mucor in the retina and choroid with neutrophilic reaction, minimal vitritis was seen in our patients on SD-OCT. Vitritis associated with retinitis is dependent upon the immune status of the patient. CMV retinitis, prevalent in immunocompromised patients is associated with minimal to complete absence of vitritis [33]. Most patients in our cohort had a history of diabetes mellitus or steroid intake associated with dampened host immune response. Thrombotic infarction of retinal arteries secondary to mucor mycosis can lead to delayed or minimized local immune response. We hypothesize this as a reason for minimal vitritis seen in our patients. There was a decrease in the number of eyes showing vitreous cells and haze at one month, with marginal increases in the intensity of both in two eyes. This followed a decline in both intensity and the number by the end of two months. A decrease in vitreous cell count and vitreous haze may suggest disease resolution or response to treatment. However, in many cases of uveitis (e.g., pars planitis), vitreous cells may persist for extended periods even after the resolution of their initiating inflammatory episode [34]. None of the eyes at final follow-up in this study showed an increase in the intensity of vitreous cell from that at one month.

In our study, 30.7% of eyes developed ill-defined focal optically empty spaces, suggestive of necrosis of retina. In one eye, these areas of necrosis were seen in the inner retina, whereas in four eyes, these areas overlaid the retinal pigment epithelium, seemingly affecting the outer retina. These focal, optically empty spaces of the outer nuclear layer are described in the literature as a part of the morphological changes seen in necrotizing viral retinitis [33]. In necrotizing viral retinitis, including herpetic retinopathy and CMV retinitis, necrosis is caused by viral-induced cytolysis and vascular occlusion. However in our study these necrotic spaces could be a spectrum of coagulative necrosis secondary to vascular occlusion by mucor. Prolonged ischemia is seen to cause cell swelling, cell rupture, and cell death by necrotic, necroptotic, apoptotic, and autophagic mechanisms [30,35]. These necrotic spaces, collapsed over time in this study, lead to further thinning of the retina. Histopathological examination of exenterated eyes in our study also revealed disruption of the normal retinal lamellar structure along with thrombotic angioinvasion (Figure 5C,D).

There was a significant reduction in retinal thickness at the end of two months for all eyes in this study. Thinned-out retina with retinal layer disorganization is suggestive of retinal atrophy. Macular atrophy with macular retinal pigment epithelial granularity is described as a late feature of retinal artery occlusion [36]. In our study, fibrous scarring of the inner retina was seen at the end of two months, more evidently in eyes that presented with the clinical picture of ophthalmic artery occlusion, and had a total retinal thickness of more than 500 µm on SD-OCT at presentation. Parapapillary fibrous scarring with RPE hyperplasia is described in the literature as a late finding in iatrogenic ophthalmic artery occlusion [37]. However, the pathophysiology associated with this fibrous scarring and SD-OCT findings is missing. RPE hyperplasia observed in this study could be attributed to choroidal infarction along with RPE cell necrosis and consequential migration of RPE cells to repopulate the region. The focal areas of chorioretinal scarring seen in this study could be sequela to a local inflammatory caused by mucor hyphae. Histopathological examination of this eye revealed mucor hyphae in the choroid with intense neutrophilic reaction (Figure 5C).

The ocular involvement in ROCM is somewhat predictable and may occur by direct invasion or hematogenous spread. Large mucor hyphae can directly invade the sclera to reach the choroid and retina or may grow along the walls of blood vessels due to their angiotropic nature. The sites at which blood vessels and nerves pierce the sclera represent potential weak points which permit direct ocular fungal invasion [5]. Infiltration of vessel internal elastic lamina by mucor commonly causes thrombotic infarction of the ophthalmic artery or its branches, mainly ciliary arteries and central retinal artery [17]. Thrombotic infarction vessels by mucor may precede the direct invasion of ocular coats due to the angiotropic nature of mucor and sclera acting as a natural barrier to invading mucor [36]. The same was evident in a patient who underwent orbital exenteration in our study on histopathological analysis; there was no evidence of mucor hyphae, although there was evidence of coagulative necrosis of retinal layers with the clinical picture of CRAO. Early infarction of the ophthalmic artery, central retinal artery, or both could be the reason for retinal infarction-like presentation in most of our study’s cases. Angiotropism by mucor of the central retinal artery and its branches could be the reason for the predilection of vitreous cells and haze seen overlaying the disc and retinal artery branches. Atrophic changes in the outer retina could also be due to coagulative necrosis caused due to ischemic infarctions. Localized chorioretinal retinal scarring seen in few eyes in this study could be a sequela to chorioretinal inflammation caused by mucor hyphae. This, along with vascular infarction, could be the reason for RPE hyperplasia in our study.

Another important observation in this study was, eleven of fifteen patients with previous COVID-19 infection history and concomitant DM in nine patients had a presenting visual acuity 20/800 or worse. Seven of nine patients in this subset had presented with no light perception visual acuity. All four eyes that underwent orbital exenteration in this study had a history of COVID-19 infection and concurrent steroids use. Four of five eyes presenting with a total retinal thickness of more than 500 µm and non-discernible inner and outer retina had a history of COVID-19, steroid use and DM, and other features such as vitreous cells, vitreous haze, thickened posterior hyaloid and RPE disruption with choroidal thinning, though seen in patients with a history of COVID-19 infection, was also seen in a patient with no history of COVID 19 infection (case 3). Uncontrolled diabetes mellitus in patients with SARS-CoV-2 infection has emerged as an important risk factor for predisposing these patients to mucormycosis. Systemic corticosteroid treatment, though seen to reduce mortality in these patients, can induce or worsen hyperglycaemia in patients with undiagnosed or pre-existing uncontrolled diabetes mellitus (DM) [3]. This development and progression of DM generates an inflammatory state which predisposes to mucormycosis [38]. The activation of antiviral immunity to SARS-CoV2 in these patients could further potentiate this inflammatory state and may cause secondary infections such as mucormycosis [39]. Cautious use of steroids and strict metabolic control can reduce the incidence of ROCM/ROM in these patients. However, further research is required to study the progression analysis of these novel OCT findings, with systemic metabolic derangements.

Our study has certain limitations. First, the small sample size and attrition due to orbital exenteration did not allow for appropriate matching, thereby affecting the power of the statistical analysis, more or less confining the result of this study to a descriptive level. Second, for most eyes, the choroid microstructure could not be analysed because of technology limitations and the non-availability of EDI-OCT. Consequently, alteration in choroidal thickness and structure could not be recorded, which could have provided insight into the site of vascular occlusion. Third, in our study, the assessment of microstructural changes was limited to the posterior vitreous and retina. OCT and histopathological assessment of the optic nerve would have been useful, as visual loss occurring in our cases can also be attributed to optic nerve ischemia seen as optic disc pallor on follow-up fundus photographs. Histopathological assessment of the optic nerve would have also provided important details about the site of vascular occlusion secondary to invasive mucormycosis. Fourth, the morbidity associated with surgical intervention in ROCM and associated complications of the disease, such as exposure to keratopathy, enophthalmos, hypotony, and others, did not allow for prolonged follow-up of these patients on SD-OCT. Lastly, as an extension to this study, FFA and ICG analysis of the patients was planned, but due to developments of renal toxicity secondary to systemic treatment and early intracranial extension of disease, serial FFA and ICG imaging was not possible for most patients. SD-OCT correlation with FFA and ICG would have been instructive but was not possible in this study. A direct comparison between OCT and histology was available for four exenterated eyes, but the results of only these patients could not represent the entire cohort in this study.

## 5. Conclusions

This novel study identifies morphological changes of the retina and vitreous on SD-OCT in eyes in vision-threatening ROCM presenting with central retinal or ophthalmic artery occlusion features. In vitro analysis of these eyes has reinforced those early morphological changes could result from retinal infarctions secondary to artery occlusion. The late changes are the possible sequelae of retinal infarction, with some contribution from the inflammation resulting from the mucor invading choroid and retina.

## Figures and Tables

**Figure 1 diagnostics-12-03098-f001:**
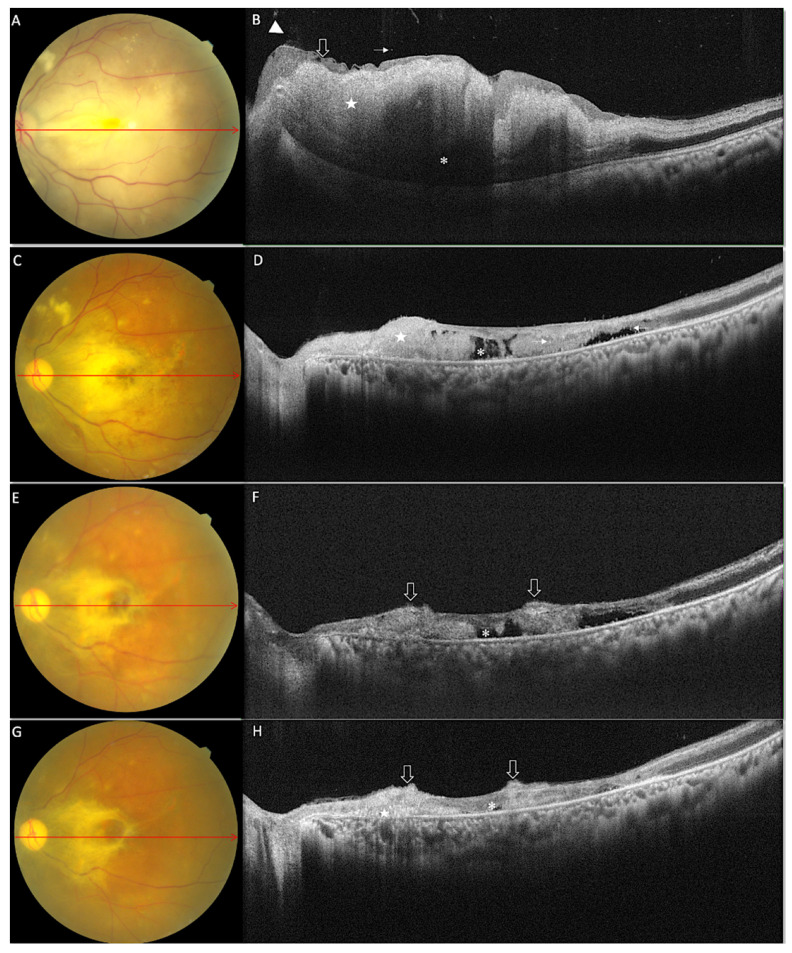
Serial fundus photographs and SD-OCT images of a representative case (case 5), at baseline (**A**) Fundus photographs showing diffuse retinal whiting with papilla-macular fold, boxcar segmentation of the vessel, absence of cherry red spot (**B**) The corresponding SD-OCT line scan shows diffuse retinal thickening, increased inner retinal hyperreflectivity (white star) with shadowing effect on the outer retina (white asterisk). Along with these retinal folds, inner limiting membrane (ILM) detachment (empty arrow), vitreous haze and vitreous cells (arrowhead) are also seen. At three weeks (**C**) fundus photographs show a reduction in the area of diffuse retinal whiting, with few cotton wool spots and few hemorrhages. (**D**) On SD-OCT disruption of all retinal layers (white star) along with optically empty spaces (white asterisk) (necrotic spaces) and hyperreflective foci (arrow head) with disruption of the photoreceptor layer and external limiting membrane. At four weeks (**E**) On fundus images, further reduction in the area of retinal whiting was observed with fibrous proliferation around the fovea. (**F**) OCT shows thinning of the retina with the collapse of optically empty spaces (white asterisk) along with a contraction of the inner retinal layer, clumping and baring of the outer retina in the para-foveal region (empty arrow). At end of eight weeks, there was (**G**) persistence of retinal whiting and fibrous proliferation on fundus photograph (**H**) with SD-OCT showing near-total collapse of optically empty spaces and a further thinning of the disorganized retina (white asterisk) (retinal atrophy). This accompanied disruption of ELM and photoreceptor layer.

**Figure 2 diagnostics-12-03098-f002:**
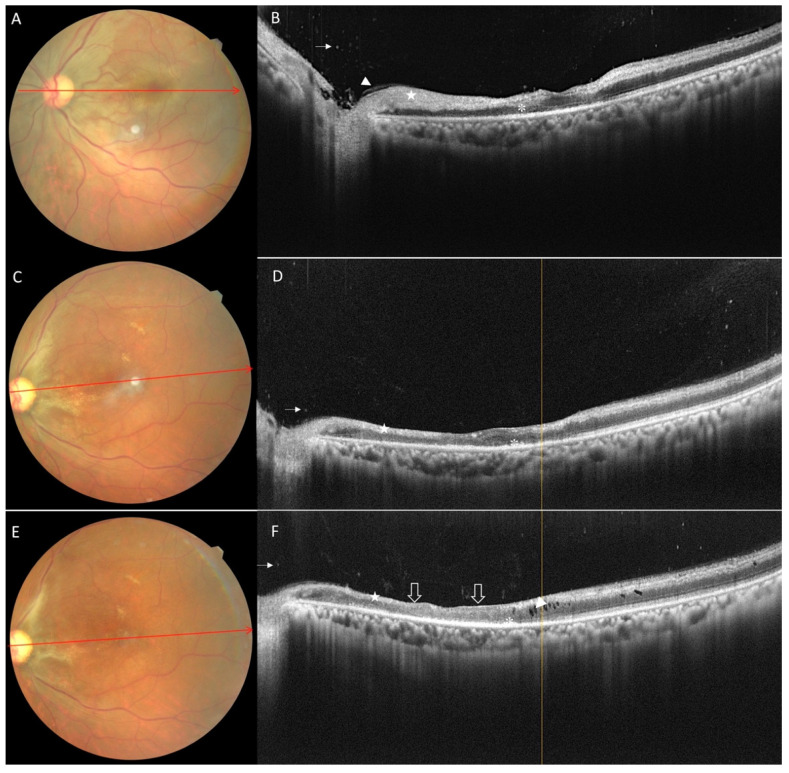
Serial fundus photographs and SD-OCT images of a patient (case 1) with delayed retinal imaging due to early debridement and associated morbidity. Imaging done at one week shows (**A**) Diffuse retinal whitening with cherry-red spot on fundus photography (**B**) the corresponding SD-OCT line scan shows increased inner retinal thickening and hyperreflectivity (white star), thinning of nasal retina at 1 mm from the fovea, vitreous cells overlaying the disc (white arrow), posterior hyaloid thickening and detachment (white arrowhead) and disruption of the photoreceptor layer and ELM (white asterisk). At the follow-up of four weeks (**C**) fundus photographs revealed improved retinal transparency with confinement of retinal opacification being to the parapapillary area. (**D**) OCT showed persistence of inner retinal hyperreflectivity, with inner as well as total retinal thinning along with disruption of the photoreceptor layer and ELM. At the eight-week follow-up (**E**) fundus picture showed marginal improvement in retinal whiting with (**F**) development of disorganized retina (area between empty arrow) on SD-OCT associated with further thinning of the retina (retinal atrophy). The inner retina, though reduced in thickness, had persistence of inner retinal hyperreflectivity (white star) with a few cystic spaces in the outer nuclear layer (white asterisk).

**Figure 3 diagnostics-12-03098-f003:**
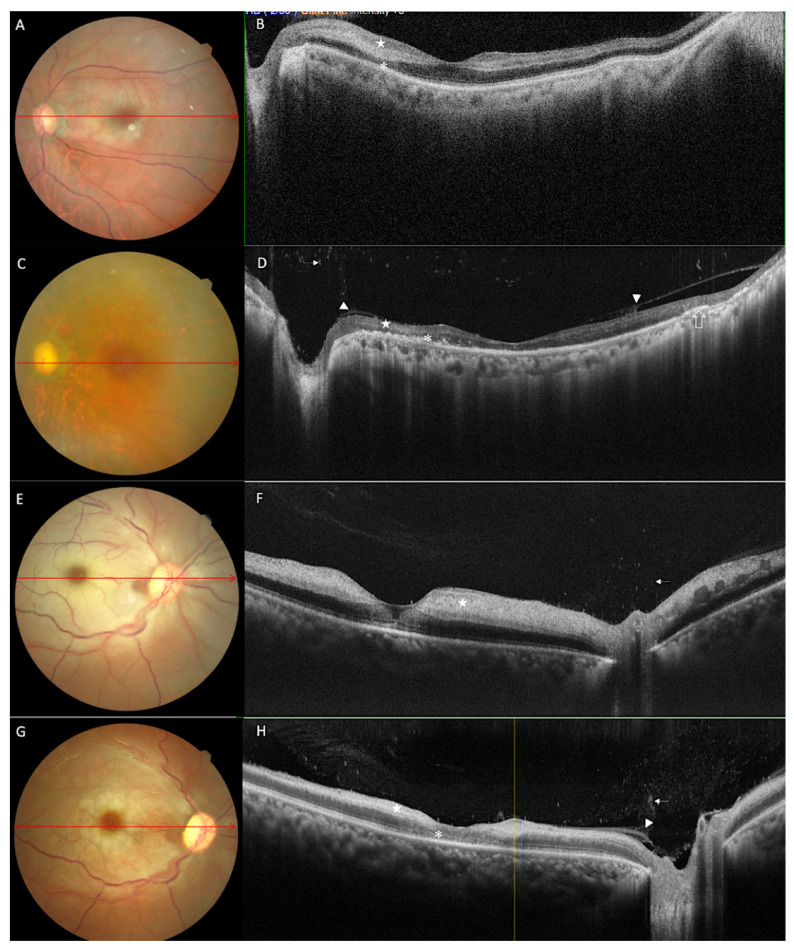
Serial fundus photographs and SD-OCT images of two representative cases: Case 10: At baseline, (**A**) fundus photographs show diffuse retinal whiting with a cherry red spot. (**B**) Morphological alterations on corresponding SD-OCT includes increased inner retinal thickness and hyperreflectivity (white star), focal disruption of ELM and photoreceptor layer with focal areas of increased reflectivity of the outer nuclear layer. On follow-up at nine weeks, (**C**) fundus photograph shows disc pallor, attenuation of superior-temporal vessels, retinal atrophy and RPE changes. (**D**) Corresponding SD-OCT reveals persistence of vitreous cells overlaying the disc (white arrow), reduction in inner retinal reflectivity, and thickness of both inner and outer retina (white star), posterior hyaloid thinking and detachment (white arrowhead), photoreceptor layer and ELM layer disruption (white asterisk) with RPE hyperplasia (empty arrow). Case 8: Imaging done at presentation shows (**E**) diffuse retinal whiting, boxcar segmentation of the vessel and cherry red spot on fundus photography. (**F**) On SD-OCT, there is increased inner retinal thickening and hyperreflectivity (white star) along with vitreous cells overlaying the disc (white arrow). At four weeks (**G**) fundus photograph reveals a mild reduction in retinal whiting with regaining of retinal transparency and (**H**) on SD-OCT persistence of vitreous cells with increased intensity along with vitreous haze, posterior hyaloid thinking and detachment (white arrowhead) persistence of inner retinal hyperreflectivity with a decrease in thickness of both inner and outer retina (white star). Fuzzy disruption of ELM and photoreceptor layer was seen in the foveal region.

**Figure 4 diagnostics-12-03098-f004:**
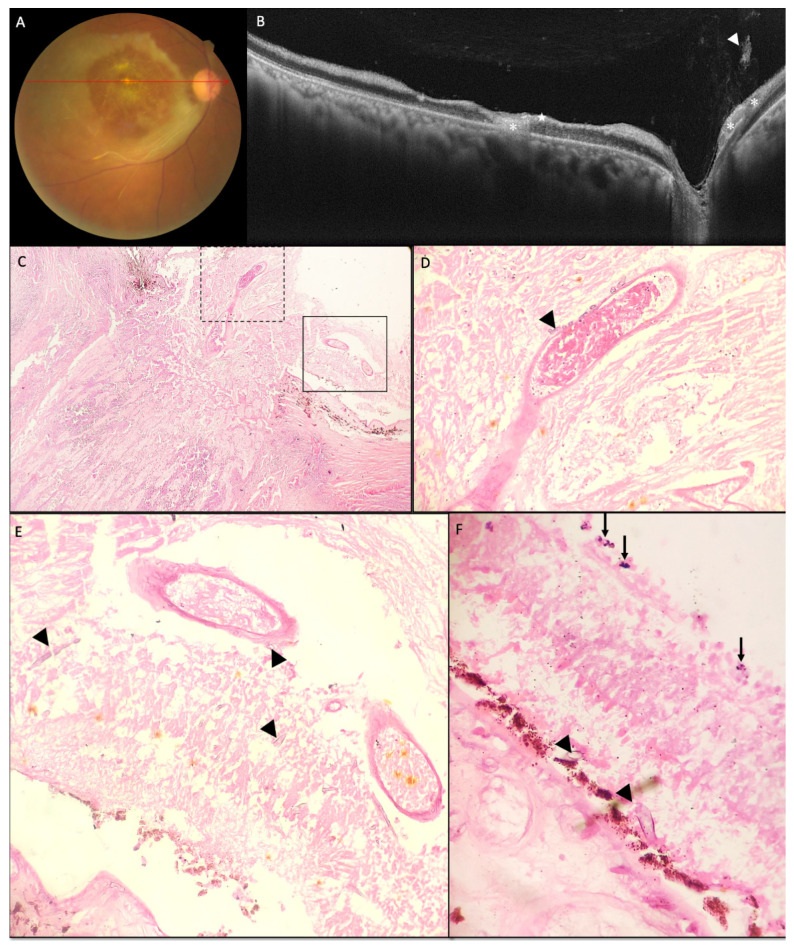
Fundus photograph, SD-OCT image and histopathological micrographs of a representative case (case 13): (**A**) Fundus photograph shows resolving retinal opacification with center clearing and inferotemporal arterial attenuation. (**B**) The corresponding SD-OCT image shows, thinned out, hyperreflective, disorganized inner retina along (white star) with outer retinal thinning along, photoreceptor layer and ELM layer disruption. Hyperreflective, disorganized retina seen at fovea suggestive of foveal scarring (white asterisk). Vitreous haze overlaying the second order vessel (white arrow head) with few vitreous cells in posterior vitreous. Photomicrograph (**C**) shows optic nerve displaying edematous and degenerative changes. (H&E; 40×). (**D**) The high-power view of the dotted line square in image (**C**) shows thrombosed blood vessel surrounded by mucor hyphae (black arrow head). (H&E; 200×). (**E**) The high-power view of the solid line square in image (**C**) shows necrotic retinal tissue displaying disruption of the normal lamellar architecture along with presence of broad aseptate ribbon-like mucor hyphae (black arrow head) in the necrotic tissue (H&E; 200×). (**F**) Photomicrograph shows retinal tissue with presence of mucor hyphae (black arrow head) in the necrotic layer of rods and cones along with neutrophils (black arrow) in the necrotic outer nuclear layer. (H&E; 400×).

**Figure 5 diagnostics-12-03098-f005:**
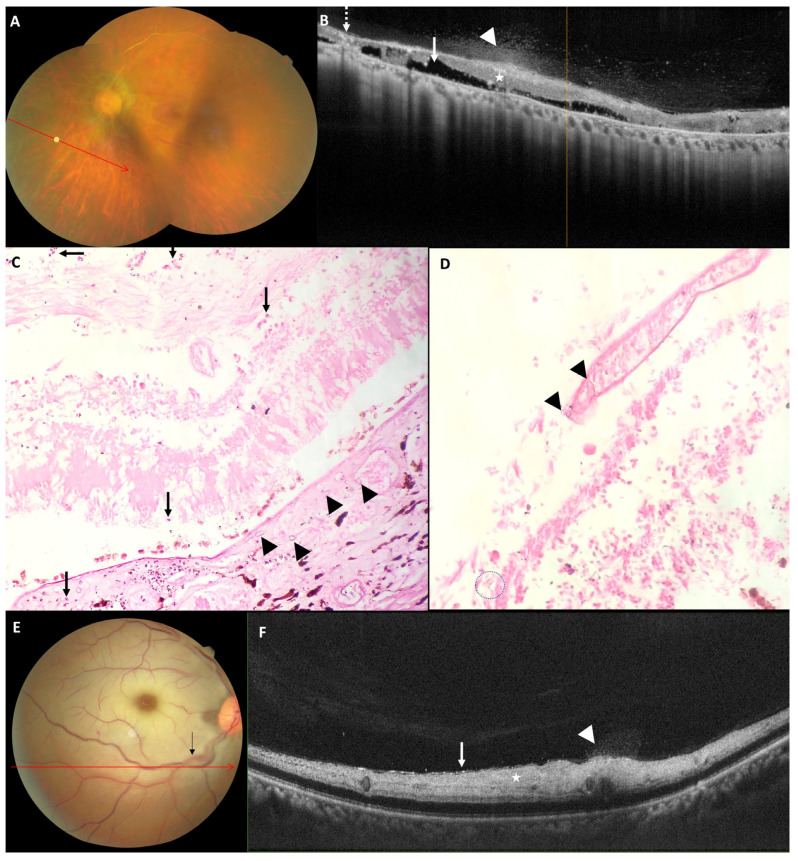
Fundus photograph, SD-OCT images having nasal vascular arcade cross-section and histopathological morphology of a representative case (case 12) at sixth week of follow up: (**A**) Colour fundus photograph montages shows sclerotic retinal arteries with boxcar segmentation, fibrous scarring (**B**) SD OCT images showing disruption of all retinal layers (white star), optically empty spaces (white arrow) (necrotic spaces), focal chorioretinal scarring (white dotted arrow) and vitreous cell and haze (white arrow head) (vitritis) overlaying thickened hyaloid. Photomicrograph shows (**C**) mucor hyphae in choroid along with angioinvasion of choroidal blood vessel (black arrowhead). Neutrophilic infiltrate (black arrow) present in choroid as well as in necrotic layer of retina (H&E; 200×). (**D**) Necrotic retinal tissue with presence of mucor hyphae (dotted circle) and disrupted normal lamellar architecture. Blood vessel showing angioinvasion by mucor hyphae (black arrowhead) (H&E; 400×). Fundus photograph, SD-OCT images of temporal vascular arcade of another representative case (case 8) at presentation (**E**) Colour fundus photograph suggestive of retinal infarction and oedema with haemorrhage along inferior temporal artery (black arrow). (**F**) With SD-OCT showing signs suggestive of retinal infarction including increased inner retinal thickness and hyperreflectivity (white star), presence of subtle internal limiting membrane detachment (white arrow), along with the sign of inflammatory response, vitreous haze and cell overlaying the vascular arcade (white arrow head), secondary to possible angioinvasion.

**Table 1 diagnostics-12-03098-t001:** Clinical and demographic features of patients with vison-threatening ROCM.

Patient No.	Age, Years	Gender	Laterality	Systemic Illness/ Condition	History of COVID-19 Infection	History of Steroid Intake	CNS Involvement at Presentation	Treatment Received
1	35	F	Unilateral	None	Yes	No	No	IV Amb + Debridement + LAMB
2	42	M	Unilateral	DM/HTN	Yes	No	No	IV Amb + Debridement + LAMB
3	22	M	Unilateral	DM Hepatomegaly Jaundice	No	No	No	IV Amb + Debridement + LAMB
4	60	M	Unilateral	DM	Yes	Yes	No	IV Amb + Debridement + LAMB
5	39	M	Unilateral	DM HTN	Yes	Yes	No	IV Amb + Debridement + LAMB
6	63	M	Unilateral	DM	Yes	No	No	IV Amb + Debridement + LAMB
7	57	F	Unilateral	DM	Yes	Yes	No	IV Amb + Debridement + LAMB
8	46	M	Unilateral	DM	Yes	No	No	IV Amb + Debridement + LAMB
9	65	M	Unilateral	DM	No	No	No	IV Amb + Debridement + LAMB
10	48	M	Unilateral	DM	No	No	No	IV Amb + Debridement + LAMB
11	40	M	Unilateral	DM	No	No	No	IV Amb + Debridement + LAMB
12	66	M	Unilateral	DM	Yes	Yes	No	IV Amb + Debridement + LAMB + Exenteration
13	32	M	Unilateral	DM	Yes	Yes	No	IV Amb + Debridement + LAMB + Exenteration
14	38	M	Unilateral	None	Yes	Yes	Yes	IV Amb + Debridement + LAMB + Exenteration
15	42	F	Unilateral *	DM	Yes	Yes	Yes	IV Amb + Debridement + LAMB + Exenteration

DM: Diabetes mellitus, HTN: Hypertension, LAMB: Local transcutaneous amphotericin B Injections, IV Amb: Intravenous Amphotericin B, * Bilateral orbital involvement with ocular involvement of one eye.

**Table 2 diagnostics-12-03098-t002:** Ocular features of eyes of patients with vison-threatening ROCM.

Patient No.	Eye	Extraocular Movements and Other Findings	Follow-Up	BCVA at Baseline	BCVA at Last Visit	*p* Value
1	LE	Restricted; Ptosis	8	20/400	20/800	0.1362
2	LE	Restricted; Ptosis; Proptosis	10	PL-ve	PL-ve	
3	RE	Restricted; Ptosis; Proptosis	9	PL-ve	PL-ve	
4	RE	Restricted; Ptosis; Proptosis	8	PL-ve	PL-ve	
5	LE	Restricted; Ptosis	8	20/800	PL-ve	
6	LE	Restricted; Ptosis; Proptosis	8	PL-ve	PL-ve	
7	RE	Restricted; Ptosis; Proptosis	8	PL-ve	PL-ve	
8	RE	Restricted; Ptosis; mild proptosis	8	20/400	20/800	
9	LE	Restricted; Ptosis	8	20/800	20/800	
10	LE	Restricted; Ptosis	9	20/400	20/400	
11	LE	Restricted; Ptosis; mild proptosis	11	HM	HM	
12	RE	Frozen globe; proptosis; Ptosis	6	PL-ve	PL-ve	
13	RE	Frozen globe; proptosis; Ptosis	7	PL-ve	PL-ve	
14	RE	Frozen globe; proptosis; Ptosis	2	PL-ve	PL-ve	
15	LE	Frozen globe; proptosis; Ptosis	1	PL-ve	PL-ve	

Follow-up duration is reported in weeks, BCVA: best-corrected visual acuity, PL-ve: no light perception, HM: Hand motion.

**Table 3 diagnostics-12-03098-t003:** Prevalence of OCT findings in patients with vison-threatening ROCM.

Features	At Baseline n (%) (n = 15)	At One Month n (%) (n = 13)	At 2 Months n (%) (n = 11)
Vitreous haze/debris	5 (33.3)	4 (30.7)	4 (36.3)
Vitreous cells	10(66.6)	9 (69.2)	7 (63.6)
Posterior hyaloid thickening with partial detachment	1 (6.6)	3 (23.07)	6 (54.54)
ILM detachment	5 (33.3)	0	0
Increased ILM reflectivity	7 (46.6)	3(23.0)	7(63.6)
Inner retinal thickening	7 (100) !	0 *	0 ^#^
Outer retinal thickening	4 (57) !	0 *	0 ^#^
Inner retinal hyperreflectivity	15 (100)	4 (66.6) *	2 (50) ^#^
Inner retinal loss of organised layer structure	13 (86.6)	6 (100) *	4 (100) ^#^
Ill-defined optically empty spaces (outer retinal tissue void)	0	4 (30.7)	3 (27.2)
Hyperreflective dots in outer nuclear layer	0	4 (66.1) *	1 (25) ^#^
Outer nuclear layer cystic spaces	0	2 (22.1) *	3 (66.6) ^#^
Disruption of external limiting membrane	4 (40.0) !	6 (100) *	4 (100) ^#^
EZ layer disruptions without hyperreflective dots	1 (10) !	6 (100) *	4 (100) ^#^
Full thickness loss of retinal architecture	0	4 (30.7)	4 (36.3)
Subretinal fluid at fovea	1 (6.6)	0	0
Inner retinal thinning	1 (6.6)	6 (100) *	4 (100) ^#^
Retinal thinning	1 (6.6)	9 (69.2)	10 (90.9)
Focal RPE disruption with choroidal thinning (scar)	0	0	3 (27.7)
Central macular thickness (mean + SD) µm	449.0 ± 308.7	194.5 ± 50.7	184.3 ± 52.1
	*p*-value = 0.0103		
Inner retinal thickness at 1 mm nasal to fovea (mean + SD) µm	168.9 ± 43.0 !	135.4 ± 21.4 *	77.34 ± 48.3 ^#^
	*p*-value = 0.0009		
Outer retinal thickness at 1 mm nasal to fovea (mean + SD) µm	198.3 ± 37.9 !	183.01 ± 18.2 *	154.87 ± 21.2 ^#^
	*p*-value = 0.0089		
Total retinal thickness at 1 mm nasal to fovea (mean + SD) µm	498.8 ± 300.2	259.1 ± 72.4	221.8 ± 69.3
	*p*-value = 0.0090		
Central macular thickness of non-involved eye (mean + SD) µm	247.26 ± 13.7	248.7 ± 13.6	245.0 ± 14.8
	^ *p*-value = 0.0144	^ *p*-value = 0.0007	^ *p*-value = 0.0003
Inner retinal thickness at 1 mm nasal to fovea of non-involved eye (mean + SD) µm	148.01 ± 9.6 !	149.56 ± 9.9 *	151.23 ± 10.9 ^#^
	^ *p*-value = 0.0636	^ *p*-value = 0.1718	^ *p*-value = 0.0036
Outer retinal thickness at 1 mm nasal to fovea of non-involved eye (mean + SD) µm	182.95 ± 11.53 !	182.97 ± 12.01 *	184.41 ± 13.03 ^#^
	^ *p*-value = 0.3162	^ *p*-value = 0.4213	^ *p*-value = 0.0478
Total retinal thickness at 1 mm nasal to fovea of non-involved eye (mean + SD) µm	330.4 ± 22	332.53 ± 22.0	331.72 ± 23.9
	^ *p*-value = 0.0401	^ *p*-value = 0.0012	^ *p*-value < 0.0001

ILM: internal limiting membrane, EZ: ellipsoid zone, RPE: retinal pigment epithelium, ! n = 7; * n = 6; ^#^ n = 4 (number of eyes with discernible inner and outer retina), ^ *p* = values depicts the level of significance of the individual mean thickness metrics of the affected eye in comparison to contralateral control eye at various time intervals.

**Table 4 diagnostics-12-03098-t004:** OCT and histopathology findings in patients undergoing orbital exenteration for vison-threatening ROCM.

Patient No.	OCT Features	Histopathology Observations
15	Increased inner retinal thickness and hyperreflectivity, focal disruption of ELM and photoreceptor layer with focal areas of increased reflectivity of the outer nuclear layer.	Focal retinal necrosis with normal choroid and optic nerve. No sign of fungal elements in retina or choroid.
14	Increased inner retinal reflectivity with thickening, thinning of nasal retina at 1 mm from the fovea, minimal vitreous cells overlaying the disc and in posterior vitreous, disruption of the photoreceptor layer and ELM.	Retina: Acute neutrophilic infiltration; areas of focal necrosis; angio-invasion of retinal vessels with thrombosis of vessel. Choroid: Areas of necrosis, presence of fungal hyphae. Optic nerve: Branch of central retinal artery showing angioinvasion (fungal hyphae in wall).
13	Trace vitreous cells with haze over retinal vessel. Disruption of inner retinal layer with inner retinal hyperreflectivity, thinning of nasal retina at 1 mm from the fovea, disruption of the photoreceptor layer and ELM.	Retina: Acute neutrophilic infiltration with necrosis in all layers with fungal hyphae above RPE layer; angio-invasion of retinal vessels. Choroid: Areas of necrosis. Optic nerve: Central retinal artery and its 1st branch showing angioinvasion (fungal hyphae in wall).
12	Vitreous cells and haze with posterior hyaloid thickening. Disruption of all retinal layers. Optically empty spaces (necrotic spaces). Multiple hyperreflective foci (arrow) with disruption of the photoreceptor layer and external limiting membrane and areas of chorioretinal scarring.	Retina: Neutrophilic infiltration, Fungal hyphae seen in ganglion cell layer, disorganised retinal lamellar structure. Choroid: Fungal hyphae with necrosis and intense neutrophilic infiltrate. Optic nerve: Oedema and inflammation.

## Data Availability

Not applicable.

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
