# Peer review of "Spectral Domain Optical Coherence Tomography Findings in Vision-Threatening Rhino-Orbital Cerebral Mucor Mycosis—A Prospective Analysis"

_diagnostics, 2022, doi:10.3390/diagnostics12123098_

Round 1

Reviewer 1 Report

The authors present a longitudinal study tracking 15 subjects with unilateral Rhino-Orbital Cerebral Mucor Mycosis treated with anti-fungal amphotericin over approximately two months using dual fundus and SD-OCT imaging on all subjects and show representative results, in addition to histological images extracted for a subset of exenterated eyes to further assess retinal structural changes. Subject co-morbidities (e.g., diabetes mellitus, jaundice, hypertension, and hepatomegaly) and history of COVID-19 infection and steroidal treatment were all reported. Statistical analysis was done by comparing quantified biomarkers relative to a patient’s initial visit and/or their contralateral eye that acted as a control.

The manuscript is well written expect for the occasional typo, improper capitalization, or naming inconsistency. I tried to highlight these as much as possible in the PDF attached, but likely missed many. Please carefully review the manuscript checking for these errors. 

Major Edits:

Interpretation of changes in ILM and inner retinal reflectivity results either relative to baseline in the diseased eye or relative to the control eye is missing. Why is this observation important? What do these changes tell us about the progression of the disease?   

The authors indicate that there is a higher prevalence of ROCM in India following the COVID-19 pandemic. The discussion is lacking how COVID-19 could put diabetic patients at higher risk for ROCM or how COVID-19 exacerbates symptoms and/or increases the rate of disease progression. 11/15 subjects had a history of COVID-19. Did subjects with a history COVID-19 experience more dramatic changes in the biomarkers quantified in Table 3 compared to the group of subjects that did not get covid?

It would be helpful to compare the authors tabulated results to the actual images in Figures 1-5. Stating the subject’s number in the figure caption goes a long way here.

P-values in Table 3 are a little messy. It’s not quite clear what thickness metrics are being used for t-tests. In the “Statistics” section please elaborate about what statistical tests were done and it could be helpful to state each p-value in the main text for clarity.  

State sampling density of your OCT scans for clinicians so those wanting to acquire similar images as reported in this manuscript can repeat the imaging protocol. The sampling density affects the image acquisition speed as well as the image resolution, so this is important to state. Also, bring the make and model information of the SD-OCT imaging device you used into the “image acquisition” section.

Why did ROCM only manifest itself unilaterally in these subjects?

Minor Edits:

The labels for Figure 5E and 5F are missing and mislabeled in the caption.

State that Table 4 is looking only at exenterated eyes. Revise main text accordingly. This wasn’t clear at first.

What is PL-ve and HMCF? Include description in Table 2 footer. Also, p-value seems to be between subjects 8 and 9, but looks like it should go to subject 8. Correct this.

Reviewer 2 Report

The manuscript entitled "Spectral Domain Optical Coherence Tomography Findings In 2 Vision Threatening Rhino-Orbital Cerebral Mucor Mycosis - A 3 Prospective Analysis" brings in the front of the readers a rare and dramatic situation of Mucor Mycosis complication. 

The following observation nave to be made:

Introduction 

Line 91- please replace "diminution" word with "decreasing".

Material and Methods 

Please insert the first paragraph  from results  in the Material and Methods chapter (except the Table 4 description).

Specify the number of the patients in the material and Methods chapter. Add the inclusion and exclusion criteria (even you have only few cases included).

Since the fundus photography showed the optic disc pallor, I consider would have been useful to include the optic nerve assessment in your OCT examination. Is there any reason you did not include it? I also consider that, in case of orbit exenteration, the examination of optic nerve, could offer very precious details about  Mucor Mycosis infection of the optic nerve.

Round 2

Reviewer 2 Report

The manuscript accomplish the conditions to be published in "Diagnosis".

Congratulations!